# Proposal for Perioperative Pharmacological Protocol for the Reduction in Early Complications in Orthorhinoseptoplasty: Five Years of Experience

**DOI:** 10.3390/jpm13091330

**Published:** 2023-08-29

**Authors:** Riccardo Nocini, Valerio Arietti, Eleonora Barausse, Lorena Torroni, Alessandro Trotolo, Giangiacomo Sanna

**Affiliations:** 1Unit of Otolaryngology, Head and Neck Department, AOUI University of Verona, 37100 Verona, Italy; 2Unit of Maxillo-Facial Surgery, Head and Neck Department, AOUI University of Verona, 37100 Verona, Italyale.trotolo@gmail.com (A.T.);; 3Unit of Epidemiology and Medical Statics, Department of Diagnostics and Public Health, AOUI University of Verona, 37100 Verona, Italy; lorena.torroni@univr.it

**Keywords:** ORSP, rhinoplasty, revision rhinoplasty, rhinoseptoplasty, therapy protocol, surgical complications

## Abstract

Septorhinoplasty is a crucial intervention in functional and aesthetic facial surgery. Although rare and usually manageable, complications could lead to disfiguring consequences. There is no universal protocol for perioperative management in the literature. The aim of this article is to analyze the surgical complications in patients who underwent open rhinoseptoplasty and were treated in the perioperative period with the standardized protocol used in our department, in order to propose it as a standardized protocol for a more global application. Methods: The patients included underwent rhinoplasty between 2017 and 2022 and were managed with the same treatment protocol. Perioperative and intraoperative data, as well as possible complications, were collected. Results: A total of 129 patients were included, 73% of which reported either mild or no complications. Abnormal scar healing was the most frequent complaint (9%), followed by edema (6.2%), nasal dyspnea (3.9%), infection (2.3%), and bleeding (2.3%). No severe complications were reported. Conclusions: Our protocol appears to be effective in minimizing complications such as infection and bleeding, although it is very difficult to compare the results with the literature.

## 1. Introduction

‘Septorhinoplasty’ is a general term used to describe a range of surgical approaches and techniques that involve manipulation of the external nasal framework and nasal septum. Broadly speaking, nasal septoplasty surgery can be divided into two categories: functional and aesthetic. Functional septorhinoplasty is indicated for nasal airway obstruction that can lead to obstructive sleep apnea, snoring, altered sensations of smell and taste, and chronic rhinosinusitis. The external and internal nasal valves, formed by the nasal septum and lateral cartilages, are critical bottlenecks for nasal airflow. Reconstructive rhinoplasty is also considered a standard treatment in selected cases of craniofacial trauma and congenital craniofacial syndromes. While in the past the preference was for functional considerations at the expense of aesthetics and vice versa, modern techniques have evolved to allow symbiosis of both goals [1]. Technically, rhinoplasty is one of the most difficult and unpredictable procedures in facial surgery because of the various anatomic components and the three-dimensional forces that the surgeon must manage during the procedure. Rhinoplasty techniques are constantly evolving to not only improve outcomes but also reduce complications. Although there is literature describing complication rates and perioperative risk factors [2,3], there remains a paucity of data relating these real-world complications to the theoretical risks described during the consent process for rhinoplasty. Fortunately, complications after rhinoplasty are relatively rare and can often be avoided. Appropriate patient selection, counseling and preoperative planning, careful technique, and meticulous postoperative care are key to successful outcomes. Surgeons must be aware of the spectrum of potential complications in order to appropriately counsel their patients, and they must be able to recognize and promptly treat them when they occur. The most commonly reported complications in the literature are perioperative bleeding, edema, ecchymosis, infection, septal hematoma, and persistent epistaxis. More serious complications such as persistent pain, deformity, anosmia, and even cerebrospinal fluid loss have been described [4,5]. Early complications usually occur within the first week after surgery: Complications related to the use of rib cartilage, soft tissue complications such as excessive edema, and other typically minor, localized, and transient problems such as acne or dermatitis, postoperative bleeding/hematoma, and infection. Bleeding-related complications such as epistaxis or hematoma are the most common complications that occur after rhinoplasty. Infections after rhinoplasty can range from mild soft tissue cellulitis to abscesses to serious complications such as brain abscess or meningitis, although this is extremely rare. Late complications usually occur in the weeks to months after surgery, often years later. These complications often occur insidiously and are not discovered until the patient’s follow-up examination. Rhinoplasty must always be performed with care, and the long-term success of the procedure must be considered for both functional and esthetic outcomes. Unfavorable scarring, patient dissatisfaction, and the need for revision are the most common late complications. Revision rates in the literature vary widely but are most commonly reported as 0% to 10% [6,7]. Technical complications include septal perforation, saddle nose deformity, open roof deformity, inverted V deformity, pollybeak deformity, and rocker deformity (these depend on technique rather than management related to osteotomy and other procedures). There are numerous clinical studies that have examined the effects of some medical applications [4,8,9], surgical techniques [10,11], and preoperative and postoperative medications [12,13,14,15] to minimize these complications, but there is not yet a standardized protocol. In the absence of standardized definitions of complications and treatment protocols, comparison of complication rates between different studies is extremely difficult, and the ability to perform high-quality meta-analyses is limited. In order to accurately characterize and quantify rhinoplasty complications, we need to develop standardized consensus definitions and treatment protocols. An important point to always keep in mind is that facial plastic surgery is unique in that complications can lead to devastating disfiguring esthetic and functional outcomes. Therefore, despite the overall low complication rates, further research is needed to mitigate the risk of poor outcomes while balancing the risks of overuse of antibiotics and other medications, including increased healthcare costs, allergic reactions, adverse drug reactions, and antibiotic resistance. The aim of this study is to analyze the early complications of rhinoplasty using specific and standardized preoperative and postoperative protocols used in our institute to improve the quality of life of patients.

## 2. Materials and Methods

A literature search was performed on Pubmed that included all articles on the topic of open septorhinoplasty. Of particular interest were articles dealing with postoperative management. A retrospective analysis was performed of patients who underwent functional rhinoseptoplasty at the Integrated University Hospital of Verona between January 2017 and December 2022 and whose postoperative treatment was performed according to the standardized protocol of our center described below. Inclusion criteria were: ORSP surgery for functional correction according to the technique described below. Patients who had undergone surgery that did not comply with the technique described below, patients who were lost to follow-up, or whose treatment duration was less than 6 months were excluded. Data are collected from the preoperative, intraoperative, and postoperative periods. As this is a proposed protocol for the management of the patient’s perioperative period, patients who were not treated according to this protocol were clearly excluded.

### 2.1. Preoperative Period

All patients undergo a high-resolution CT scan for proper preoperative planning. A preoperative photographic assessment documents the patient’s preoperative status. Preoperative photographs with static and dynamic images in frontal, three-quarter, lateral, base (worm’s eye), and dorsal (bird’s eye) views should adequately document the position and shape of the nose, facial asymmetries, and the effects of smiling on the nose. Imaging can help demonstrate proposed surgical outcomes that are otherwise difficult to verbalize, improve communication between patient and surgeon, and set realistic expectations for surgery. A rhinendoscopic examination can help analyze the narrowest sites and any septal spurs or abnormalities that need to be corrected, as well as the status of the nasal valve and whether there is hypertrophy of the inferior turbinate. A medical examination must be performed, including preoperative risk stratification and medical optimization. A careful history is aimed at predicting any problems in the perioperative course. Surgery is performed under general anesthesia via orotracheal intubation and under complete intravenous anesthesia. Preoperative prophylaxis with amoxicillin/clavulanic acid 2 g iv is administered 1 h before surgery (clarithromycin 500 mg is used 1 h before if the patient is allergic to penicillin).

### 2.2. Surgical Procedure

General anesthesia or intravenous techniques (e.g., propofol) are used for open rhinoplasty. A single dose of an intravenous antibiotic covering the skin flora must be administered preoperatively. A single dose of intravenous steroid injection (e.g., 8 mg dexamethasone) may help with swelling. Local anesthetic (a 1:1 mixture of 1% lidocaine with 1:100,000 epinephrine and 0.5% bupivacaine with 1:200,000 epinephrine) is infiltrated along the septum, columella, margin, soft tissue triangle, lateral walls, and dorsum, taking care not to disfigure the appearance of the nose by excessive injection. Nasal plugs soaked with decongestant nasal spray are inserted into the bilateral nasal cavity. The vibrissae are trimmed to facilitate visualization and reduce postoperative crusting. A topical antiseptic, such as povidone–iodine dye, is used to disinfect the patient’s face.

Important maneuvers in performing OSRP are:I.Opening the nose through a mid-columellar inverted V incision, which should be placed where the underlying cartilage is closest to the skin to avoid visible scarring and contracture. Using skin hooks and a sharp dissection, the midcolumellar incision is transferred to the marginal incisions, taking care not to injure the medial and lateral crura. The soft tissue envelope is retracted further upward into a relatively avascular supra-perichondrial plane to expose the superior lateral cartilage (ULC). At this point, the dissection above the ULC is moved to a subperichondrial level with a sharp dissection. Then, using a periosteal elevator, elevate the periosteum above the nasal bone to the nasofrontal angle. The inferior lateral cartilages are then cut in the midline to expose the anterior septal angle (ASA) and prepare for septoplasty and/or septal cartilage harvest.II.Septoplasty: the ASA is sharply exposed and a sub mucoperichondrial pocket is created caudally to the nasal spine and posteriorly over the bony-cartilaginous junction of the septum on both sides. If the surgeon provides spreader grafts, the upper lateral cartilage can be sharply detached from the dorsal nasal septum to the nasal bones. At this point, the septal cartilage can be harvested, taking care to leave a 1.5 cm L-strut to provide adequate structural support. Bony deviations and/or spurs are removed according to the principles of septoplasty.III.The surgeon can then proceed with rhinoplasty, which consists of several precise maneuvers depending on what the corrections require, which can be summarized in the following points: reduction of the dorsal cusp, spreading of grafts or flaps, stabilization of the base of the nose, contouring of the nasal tip, dorsal augmentation, grafts of the nasal wing, reduction of the base of the nasal wing, and osteotomies.IV.Closure: a septal splint (e.g., a cut silicone sheet) can be sutured in place. However, this is optional as long as adequate coagulation of the septum has been performed. The transcolumellar incision is then closed with either interrupted permanent sutures (6-0 or 7-0 nylon) or fast-absorbing sutures, the latter offering a similar esthetic result without the discomfort associated with suture removal. Marginal incisions are closed with interrupted absorbable sutures (5-0 almost).V.Taping and banding: A flesh-colored tape is carefully applied from the nasofrontal angle to the supratip to reduce postoperative edema. A longer strip is placed around the infraspinous tip lobe and acts as a sling to keep the tip in the intended level of rotation. A nasal dressing is then applied with a thermoplastic splint material that becomes soft and pliable on contact with hot water and hardens on cooling. Non-absorbable nasal fillings (Merocel ^®,^, Minneapolis, MN) with antibiotic ointment are placed in both nostrils and a mustache bandage is applied.

### 2.3. Postoperative Period

After the surgical procedure, antibiotic prophylaxis is continued for the next 48 h, and amoxicillin/clavulanic acid is administered every 8 h. The patient remains hospitalized for 2 nights. Methylprednisolone 125 mg iv is given to reduce swelling: 3 times daily on the day of surgery, then 2 times on the day after surgery, and 1 time 2 days after surgery. An antireflux agent is also administered: pantoprazole 40 mg iv: 1/day. For pain control, 1 g of paracetamol is given every 8 h, and if the pain is uncontrollable, ketoprofen 160 mg iv can also be given. The nasal packing is removed the day after surgery. At discharge, the patient must follow precise instructions to maximize recovery. Antibiotic administration must be continued for 6 days (amoxicillin/clavulanate 1 gr every 8 h for 6 days). Analgesics will be administered only as needed. Nasal rinses with isotonic saline are mandatory 2–4 times a day for 14 days. After rinsing, the patient must use 1 spray of mometasone per nostril and then a decongestant spray (e.g., NTR) 1 spray per nostril for 4 days 10 min after the mometasone spray. Antibiotic ointment (fusidic acid or gentamicin): applied to columella and nostrils 3 times daily for 10 days. Finally, optional: Kelairon ointment to reduce hematoma 5 times daily until absorption. Apart from the drug treatment, the patient must follow precise instructions, such as wearing a nasal cast for 7 days after discharge and not wearing glasses for 1 month, with instructions to use lenses if possible; no blowing the nose for at least 1 month; and sneezing with the mouth open to avoid pressure trauma. The patient may bathe, but only from the neck down, taking care not to get the nasal medication wet. The patient should also avoid physical exertion for 2 weeks, avoid hot temperatures and dusty areas, not smoke, and avoid facial sun for at least 3 months, using a 50+ sunscreen if unavoidable. The protocol is summarized in Table 1.

## 3. Results

A total of 129 patients were included in the study. They were divided into 75 men and 54 women. Open septorhinoplasty was performed in the majority of cases for posttraumatic correction of deformities and respiratory problems, accounting for 52 of 129 procedures (40.3%). Another large proportion of indications was the functional correction of nasal breathing problems. In 14 patients, this was the final step in the correction of labiopalatoschisis. Revision surgery was performed in 20 patients and 3 for other reasons (e.g., aesthetic). Regarding complications, 92 patients (71.3%) had neither a mild nor a severe complication. The most important complication was scarring of the surgical wound, which was perceived by the patients as unaesthetic (even if minor), which was noted in 12 patients (9.3%). Interestingly, 11 of these patients were the result of revision surgery. Persistent edema was noted in 8 patients (6.2%). As for the most commonly described surgical complications, infection was noted in only 3 patients (2.3%) and was treated with more aggressive treatment with iv antibiotics. Excessive bleeding was observed in only 2 patients (1.6%). Other complications included nasal dyspnea in 5 patients (3.9%), paresthesia in 2 patients (1.6%), anosmia in 1 patient (0.8%), persistent algia in the operated region in 3 patients (2.3%), and other unspecified and subjective symptoms in 1 patient. A total of 102 patients required no revision surgery (79.1%), whereas 25 patients (19.4%) required a single revision and only 2 patients (1.6%) required multiple revisions. Of the 26 patients who required revision surgery, 15 patients were satisfied and had no complications, while 11 patients had inadequate wound healing (as described above). Other unspecified and subjective symptoms in 1 patient. It is important to note that the success rate for surgery of labiopalatoschisis-associated deformities is very low, approximately 28.57%. In these patients, the complication rate is 71.43%, as shown in Table 2 and Table 3.

## 4. Discussion

The purpose of this article is to propose a standardized protocol for the perioperative management of patients with OSRP, which is lacking in the literature. Surgical techniques are discussed in detail in the literature and are not the subject of this paper. Therefore, we performed a retrospective analysis of our patients in whom this procedure was performed for functional purposes. The first thing to notice is that we have a complication rate of 28.7%, which is high compared to the literature. This is due to our aim to increase the sensitivity as much as possible by considering complications, even minor events, that other authors consider normal. This could also be considered a limitation of the study, as it is not always possible to accurately quantify and standardize complications. This is consistent with what we find in the literature, as there are no standardized scales to assess these complications. Our protocol may seem like overtreatment, considering that we systematically use nasal packs and antibiotic profilaxis. We prefer this because complications in facial plastic surgery carry the risk of significant morbidity and cost, considering the potentially disfiguring events. The face is the most important area of nonverbal communication when it comes to showing emotion and facilitating social interactions. Facial disfigurement can have profound psychosocial effects, including altered body image, decreased quality of life, and low self-esteem. The most commonly reported difficulties are related to negative self-perception and impaired social interaction. No complications occurred in 92 patients (71.3%), a consistent rate. Although the use of nasal packing is not mandatory after septoplasty, we prefer it. The literature shows that the sole use of a transseptal suture after septoplasty without the use of a nasal packing has a similar success and causes less pain and discomfort for the patient [15]. Considering that bleeding may require revision surgery or repositioning of nasal packing in some cases, which may lead to suboptimal results with scarring and inadequate wound healing, we prefer the use of packing. In the literature, the bleeding rate is reported to be 0.2% to 6.7% [2,6,7]. In our case, only two patients experienced mild bleeding that did not require revision surgery. This was not affected by the use of ketoprofen for pain management. Regarding infections, the most recent 2017 CDC guidelines recommend the administration of antimicrobial prophylaxis prior to skin incision for clean and clean-contaminated procedures, based on published clinical practice guidelines. However, additional prophylactic administration of antimicrobials is not recommended, even when surgical drainage is present [16]. Most studies investigating perioperative antibiotics have shown no additional benefit [17,18]. Infection rates after rhinoplasty have been shown to range from 0 to 15%, while most authors report 2% or less [17,19,20]. However, the heterogeneity of these study populations, as well as the small sample sizes and lack of standardized perioperative care, prevent extrapolation of these data. Therefore, although postoperative infections are rare, we prefer to routinely administer prophylactic antimicrobial therapy. A recent clinical practice guideline published by the American Academy of Otolaryngology–Head and Neck Surgery (AAO-HNS) states that selective use of antibiotics is recommended in cases such as “revision surgery, complicated rhinoplasty, the use of nasal implants, extensive cartilage grafting, or in immunocompromised patients in whom the risk of infection and [resulting] morbidity outweigh the risks of antibiotic administration” [21]. Our current practice is to administer a single preoperative dose of clarithromycin if the patient is allergic to penicillin and to continue prophylaxis for one week with amoxicillin/clavulanate or clarithromycin. In our cases, we report an infection rate of 2.3% (3 of 129 patients), which is consistent with the literature. The use of prophylactic antibiotics in facial plastic surgery remains highly controversial. This is mainly because there is a lack of high-quality research and therefore it is not possible to make a convincing argument for or against the use of antibiotics. Facial plastic surgery is unique, however, in that surgical site infections can lead to devastating aesthetic and functional outcomes, such as a saddle nose after rhinoplasty or skin necrosis after facial rejuvenation. Therefore, despite overall low infection rates, further research is needed to reduce the risk of poor outcomes while offsetting the risks of antibiotic overuse, including increased healthcare costs, allergic reactions, drug resistance, and opportunistic infections. Regarding postoperative edema, Kargı et al. have shown that preoperative and intraoperative (as well as on postoperative days 1 and 2) intravenous administration of 8 mg dexamethasone reduces edema and ecchymosis [22]. Studies show that preoperative administration of steroids reduces edema and ecchymosis and that repeated administration of steroids was more effective than administration of a single dose [23,24]. Our results are consistent with the literature. In our reports, only eight patients had persistent edema with slow resorption at 1 month. Our protocol includes intraoperative and postoperative administration of steroids. In the perioperative course, we administer methylprednisolone 125 mg iv for the next three days after surgery. Studies have shown that applying a strip after rhinoplasty causes compression on the osteochondral scaffold under the skin and on the subcutaneous tissue and prevents extravasation into the subcutaneous tissue, reducing edema [25]. Hoefflin has shown that application of strips to the nose reduces edema by 4–6 weeks in the postoperative period [26]. Our protocol also includes the application of strips and plaster casts to the nose. Studies have demonstrated the psychological benefit of applying strips to the nose and periorbital area as well as the physiological benefit. The postoperative reduction in edema and ecchymosis has a positive effect on patients’ expectations of surgery, as well as on their perceptions and psychological state [9]. Several other drugs have been reported in the literature to reduce edema and improve outcomes after surgery, but none of them have been standardized and require further study before being introduced into routine clinical practice. Szagar et al. have shown that the use of oral isotretinoin after rhinoplasty in patients with thick nasal skin significantly accelerates the improvement of cosmetic outcomes in the first months after surgery, although it does not significantly affect the final cosmetic outcome 1 year after surgery [9]. Many studies investigate the use of tranexamic acid (TXA) to reduce periorbital edema and ecchymosis in open rhinoplasty and show a positive effect in this regard [27,28]. The existing literature investigating TXA in facial aesthetic plastic surgery is sparse, with varying levels of evidence and heterogeneous data. TXA may reduce postoperative edema and/or ecchymosis in rhinoplasty, although the lack of validated rating scales does not provide sufficient evidence to support this claim [29]. Topical and subcutaneously injected TXA are new routes of administration. As the results are often inconclusive and the studies are of poor quality, we do not use tranexamic acid in the postoperative course but do not exclude the possibility that it may be a valid drug in the future. Other treatments and medications such as the use of decongestants and other topical ointments (e.g., Kelairon^®^, a topical antioxidant cream used to accelerate resorption of hematomas) are based on our experience and require further studies to prove their true efficacy.

## 5. Conclusions

Our protocol seems to be effective in minimizing complications such as infection and bleeding, although it is very difficult to compare the results with the literature. There are multiple studies in the literature for each intervention that make up the protocol, but there are several biases because it is not always possible to extrapolate the benefit of each element. For this reason, we propose to standardize the protocols to better evaluate and standardize the treatment of patients undergoing surgery with OSRP worldwide.

## Figures and Tables

**Table 1 jpm-13-01330-t001:** Our protocol for perioperative management of open septorhinoplasty.

Perioperative Management Protocol for Open Septorhinoplasty			
	Preoperative	Postoperative	Post Discharge Home Care
**Pharmacologic**	Amoxicillin + clavulanic acid 2 g iv: 1 h before surgery	Amoxicillin + ac clavulanic acid 1 g iv: every 8 h Methylprednisolone 125 mg iv: day of surgery: 3/day; 1st gpo: 2/day; 2nd gpo: 1/day Pantoprazole 40 mg iv: 1/day Paracetamol 1 g iv: every 8 hrs Ketoprofen 160 mg iv: as needed	Amoxicillin + Ac. clavulanic acid: 1 g every 8 h for 6 days Analgesics as needed (e.g., acetaminophen 1 g, maximum 3 times daily or ibuprofen 400 mg, maximum 2 times daily) Nasal rinses with isotonic solution (seawater) 2–4 times daily for 14 days Nasal spray with mometasone (e.g., Nasonex spray or similar): 1 spray per nostril 2 times daily (morning and evening) for 10 days after nasal rinsing Decongestant nasal spray (e.g., NTR nasal spray or similar): 1 spray per nostril 2 times daily for 4 days, 10 min after using nasal spray with mometasone Antibiotic cream (fusidic acid/gentamicin): apply to nostrils and nasal vestibule 3 times daily for 10 days Optional: Kelairon ointment to reduce the hematoma: 5 applications/day in the area of the hematoma
**Indications**		Removal of nasal swabs on average on the first postoperative day	Nasal protection is maintained until further notice (average 7 days) Showering is allowed from the neck down (avoid high temperatures), taking care not to get the nasal dressing wet Avoiding heavy blowing for at least one month Sneezing with the mouth open for 20 days Avoiding physical exertion for 2 weeks Wearing glasses/sunglasses is not allowed/recommended for 1 month, contact lenses can be used Avoidance of excessive temperature fluctuations, dusty environments, and situations with risk of facial trauma Avoidance of direct sun exposure and application of maximum sunscreen (SPF 50+) to the face for at least 3 months if unavoidable Absolute abstinence from smoking

**Table 2 jpm-13-01330-t002:** Population of this study.

	Summary
	(N = 129)
**INDICATIONS**	
Post-traumatic	52 (40.3%)
Functional	40 (31.0%)
Labiopalatoschisis	14 (10.9%)
Revision surgery	20 (15.5%)
Other	3 (2.3%)
**COMPLICATIONS**	
No	92 (71.3%)
Nasal dyspnea	5 (3.9%)
Infection	3 (2.3%)
Abnormal wound healing	12 (9.3%)
Paresthesia	2 (1.6%)
Anosmia	1 (0.8%)
Persistent edema	8 (6.2%)
Other	1 (0.8%)
Bleeding	2 (1.6%)
Persistent pain	3 (2.3%)
**REVSION SURGERY**	
No	102 (79.1%)
Single	25 (19.4%)
Multiple	2 (1.6%)
**COMPLICATIONS IN REVISION**	
No	15 (55.6%)
Abnormal wound healing	11 (40.7%)
Other	1 (3.7%)

Total sample: N = 129.

**Table 3 jpm-13-01330-t003:** Association between indications of surgery and complication.

Indication	Any Complication	
	No (n = 92)	Yes (n = 37)
Posttraumatic	39 (75.0)	13 (25.0)
Functional	31 (77.5))	9 (22.5)
LPS	4 (28.6)	10 (71.4)
Revision	16 (80.0)	4 (20.0)
Other	2 (66.7)	1 (33.3)

## Data Availability

Data can be found in the computerized archive of Policlinico G.B. Rossi, AOUI University of Verona. Piazzale Luduvico Antonio Scuro, 10, 37134 Verona VR, Italy.

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
