# Peer review of "Proposal for Perioperative Pharmacological Protocol for the Reduction in Early Complications in Orthorhinoseptoplasty: Five Years of Experience"

_jpm, 2023, doi:10.3390/jpm13091330_

Round 1
Reviewer 1 Report
Dear authors,
The article is very well written, reflecting a real clinical experience. The methodology is rigorous, with well-defined inclusion and exclusion criteria. The results are explicitly formulated and realistically compared with relevant results from the literature. The perioperative pharmacological protocol is very clearly described, along with the non-pharmacological protocol. The discussion section is very practical, with direct implications in current practice, bringing arguments for an apparent overtreatment, in the idea of ​​preventing early postoperative complications after ORSP.
The article is written in clear scientific language. References are relevant.
I find the article valuable as a reference for the perioperative pharmacologic protocol in ORSP, I congratulate you on your work and I recommend the manuscript for publishing with a few minor revisions that I suggest below:
- Line 38: both goals [1], instead of ”both goals[1]”
- Line 196: I suggest (e.g. aesthetic) instead of ”(aesthetic, …)”
- Line 283: The phrase ”reduces edema by 4-6 in the post-operative weeks [27]” seems to miss a word after ”4-6”
- Line 319: it is not filled with text, maybe it is necessary to delete the line.
Author Response
Dear Reviewer,
We are very pleased with your appreciation of the article. We sincerely thank you for your review and suggestions.
Regarding revisions:
Line 38: corrected.
Line 196: corrected
Line 283: corrected, one word was missing, as you have informed us
Line 319: corrected
With best regards,
The Authors.
Reviewer 2 Report
It is an important research topic. However, I have some suggestions and corrections to the article that are appended below.
Point 1: There is a need to rewrite the abstract. Rational of the study should be written clearly.
Point 2: What is more extensive use?
Point 3: Abstract: The conclusion should be clear and specific.
Point 4: Material Methods: Describe the proposed protocol for the management of patients.
Point 5: Material Methods: Subjects of the study should be discussed in this section.
Point 6: Discuss the study protocol and previous studies' protocol. Is there any advantage of this protocol over others?
Point 7:This study has not concluded about the use of antibiotics.
Author Response
Dear Reviewer,
Thank you very much for your evaluation and suggestions. We have made the appropriate corrections to make the article as clear and valuable as possible.
Regarding revisions:
Point 1 and 2: We have rewritten the abstract to make the main rationale of the study clearer.
Point 3: The conclusion in the summary has now been rewritten and should be clearer.
Ponit 4 and 5: our protocol is divided into preoperative management, surgical procedure and postoperative. All measures described in Material and Methods refer to our management. The protocol is also summarized in table 1
Point 6: As mentioned in the article, there are no standardized protocols in the literature. Therefore, our goal is to propose this protocol to try to standardize the perioperative management of patients undergoing OSRP worldwide.
Point 7: In line 253, we have stated our position on the use of antibiotics, although it is not entirely clear in the literature. “Therefore, although postoperative infections are rare, we prefer to routinely administer prophylactic antimicrobial therapy.”
With best regards,
The Authors.